# Zoopharmacology: A Way to Discover New Cancer Treatments

**DOI:** 10.3390/biom10060817

**Published:** 2020-05-26

**Authors:** Eva María Domínguez-Martín, Joana Tavares, Patrícia Ríjo, Ana María Díaz-Lanza

**Affiliations:** 1CBIOS-Center for Research in Biosciences & Health Technologies, Universidade Lusófona de Humanidades e Tecnologías, Campo Grande 376, 1749-024 Lisbon, Portugal; evam.dominguez@uah.es (E.M.D.-M.); joanatavares@hotmail.com (J.T.); patricia.rijo@ulusofona.pt (P.R.); 2Department of Biomedical Sciences, Faculty of Pharmacy, University of Alcalá, Carretera Madrid-Barcelona, Km 33.100, 28805 Alcalá de Henares, Madrid, Spain; 3Instituto de Investigação do Medicamento (iMed.ULisboa), Faculdade de Farmácia da Universidade de Lisboa, Av. Prof. Gama Pinto, 1649-003 Lisbon, Portugal

**Keywords:** zoopharmacology, zoopharmacognosy, zootherapy, cancer, multidrug resistance (MDR), collateral sensitivity, oncotherapy

## Abstract

Zoopharmacognosy is the multidisciplinary approach of the self-medication behavior of many kinds of animals. Recent studies showed the presence of antitumoral secondary metabolites in some of the plants employed by animals and their use for the same therapeutic purposes in humans. Other related and sometimes confused term is Zootherapy, which consists on the employment of animal parts and/or their by-products such as toxins, venoms, etc., to treat different human ailments. Therefore, the aim of this work is to provide a brief insight for the use of Zoopharmacology (comprising Zoopharmacognosy and Zootherapy) as new paths to discover drugs studying animal behavior and/or using compounds derived from animals. This work is focused on the approaches related to cancer, in order to propose a new promising line of research to overcome multidrug resistance (MDR). This novel subject will encourage the use of new alternative prospective ways to find new medicines.

## 1. Introduction

Cancer comprises a group of diseases characterized by abnormal cells grown without control, with the potential to invade and to spread through the body [1]. Nowadays, although all the therapeutic options available (surgery, radiotherapy, monotherapy, and polytherapy combination regimens that can include the simultaneous use of biological drugs) can reduce tumor size and increase life expectancy [2,3], cancer is still considered one of the main causes of morbidity and mortality worldwide according to the World Health Organization (WHO) [4]. This is not only due to new cases or relapses but especially because tumors are gained cross-resistance to several unrelated chemotherapeutic agents, developing a multidrug resistance (MDR) phenotype, ultimately remain the second cause of death in developed countries [3,5]. 

Recent evidence points toward stem-like phenotypes in cancer cells, promoted by cancer stem cells (CSCs), as the main culprit of cancer relapse, resistance to radiotherapy, hormone therapy, and/or chemotherapy, and metastasis. Many mechanisms have been proposed for CSC resistance, such as drug efflux through ABC transporters, microenvironment modulation, epigenome, exomes, overactivation of the DNA damage response, apoptosis evasion, increased unfolded protein response (transforming the cells particularly susceptible to endoplasmic reticulum stress and mitochondrial damage), autophagy deregulation, metabolic alterations, prosurvival pathways activation, and/or cell cycle promotion [3]. These mechanisms are still not completely understood and could occur simultaneously [6]. Nonetheless, targeted therapy toward these specific CSC mechanisms is only partially effective to prevent or abolish resistance, suggesting underlying additional causes for CSC resilience [3]. 

This resistance can be (a) intrinsic or (b) acquired, being the former caused by pre-existing factors of the tumor that are present before any treatment administered, making consequently certain treatments useless at non-toxic doses. The latter appears to be a relatively common issue throughout the administration of treatment and seems to be the main perpetrator of treatment failure in cancer patients, usually after a relapse. Increasing evidence demonstrates that cells with acquired resistance to a specific drug are prone to exhibit cross-resistance to other chemotherapeutics. This implies the existence of common mechanisms of resistance, which may be independent of the particular action of the chemotherapeutic agent. Importantly, clinical evidence shows that phenotypes of resistance can be reverted to a sensitive phenotype and suggests that cancer-associated genetic alterations are not the only players in resistance. Cancer treatments should target not only the resistance mechanisms already present in the bulk of cancer cells but also those activated in CSCs [3].

The influence of genetics and the environment on cell evolution was suggested to affect individual cells at various levels. Modulation of the CSC epigenome, with the acquisition of undifferentiated, pluripotent, and drug-resistant phenotypes, occurs preferably when CSCs are exposed to environmental stressors, such as inflammation, toxic compounds (including drugs), and/or radiation [3]. It may be possible that some nuclear medicine cancer diagnostic and therapeutic techniques favoring autophagy resulting in the development of resistance [7].

To combat drug resistance, there are two main strategies: (1) Drugs with novel modes of action to bypass resistance to established drugs or (2) inhibitors of resistance mechanisms for resensitization of tumor cells [8]. The most general approach has been the development of P-glycoprotein (P-gp) inhibitors to co-administer with anticancer drugs [6]. This consists of the pharmacological blockage of drug transporters such as P-gp (a type of ABC transporter) [8], that can lower intracellular drug concentration by expelling the drug from cancer cells [5]. However, despite their great in vitro success, there is no P-gp inhibitor currently available for clinical use [6].

Therefore, novel forms of treatment are seeking and have been purposed to overcome this problem, such as the use of hybrid combinations that consist of the association of anticancer drugs with bioactive phytochemical constituents of plant extracts [2]. Recently, it has been hypothesized that natural products, such as marine drugs, may deliver promising lead compounds for the development of collateral sensitive anticancer drugs. The phenomenon of collateral sensitivity consists of the hypersensitive to specific drugs by tumors with cross-resistance to numerous cytostatic drugs [8]. Although many hypotheses have been proposed, the mechanism of collateral sensitizing compounds remains unclear; nonetheless, they seem to be related to diverse biochemical mechanisms. A recent work assessed the ability of the macrocyclic diterpene plant derivatives as collateral sensitizing compounds in human tumor gastric (EPG85-257), pancreatic (EPP85-181), and colon (HT-29) cell models (drug-sensitive and drug-resistant sublines) [6].

These recent therapeutic proposals evidence of the continuing natural products playing a role in drug discovery and development [9]. A comprehensive review of human drugs introduced since 1981 suggests that, from 847 small molecule-based drugs, 43 were natural products, 232 were derived from natural products (usually semi-synthetic), and 572 were synthetic molecules. However, 262 of the 572 synthetic molecules had a natural product-inspired pharmacophore or could be considered natural product analogues [10]. Besides, more than 60% of current anticancer drugs have their origin from natural sources [9].

The study of natural products has been named as Pharmacognosy, which is the multidisciplinary science that studies the physical, chemical, biochemical, and biological properties of natural drugs and drug substances from plants, animals, fungi, and microorganisms for new drug discoveries [11].

Historically, natural medicines have been used to enhance human and veterinary health since immemorial times as it has been compiled in ancient tales, scriptures among other historical literature. All these texts claim the use of natural remedies to solve first health troubles in different parts of the world [11].

One of the potential areas of research in the field of Pharmacognosy is Ethnopharmacology [11]. This term coined in 1967, consists of the scientific approach to study the biological activities (beneficial or toxic effects) of any traditional preparation used by humans, using observation, description, and experimental research. Nowadays, this area has been greatly expanded, covering a wide range of topics based on anthropological, historical, and other socio-cultural studies [12].

In this regard, although plant-derived products have dominated human pharmacopeias for thousands of years, it also appeared that people from traditional societies learned from animals about the medicinal value of some plants that may not otherwise have been considered to be medicinal. This field is known as Zoopharmacognosy (Figure 1) and consists of investigating the self-medication behaviors of animals [13].

Furthermore, the Materia Medica of animals (Zootherapy, Figure 1), which is the use of animal compounds by humans for medicinal purposes, have been collected in different sources [11].

In this paper, it is proposed for the first time the Zoopharmacology (comprising Zoopharmacognosy and Zootherapy) as an encouraging way to find new cancer treatments in the context of helping to overcome MDR inhibiting drug transporters and/or by new drugs that hypersensitize resistant cancer cells through collateral sensitivity (Figure 2). Different concepts related to these areas are going to be explained following by a critical discussion of this topic. 

## 2. Zoopharmacognosy

In the plant world, a common line of defense is to produce a variety of toxic secondary compounds such as sesquiterpenes, alkaloids, and saponins which prevent predation by animals [14]. Early in the co-evolution of plant-animal relationships, some arthropod species began to utilize these chemical defenses of plants to protect themselves from their predators and parasites. It is likely, therefore, that the origins of herbal medicine have their roots deep within the animal kingdom [15].

Different species of animals seek and use the organic and inorganic substances they find in their environment to enhance their health, as it has been observed by field researchers [16,17]. This self-medication behavior was called Zoopharmacognosy. Although evidence for self-medication in non-human animals was initially mostly anecdotal, increased research in this area over the past two decades has resulted in convincing evidence of self-medication in several vertebrates and invertebrates [18]. Zoopharmacognosy has only been considered a legitimate scientific discipline since 1987, even though the recently published paper by Mezcua et al. [17] highlighted that this discipline goes back as least as Ancient Greece.

The basic premise of Zoopharmacognosy is that animals utilize plant secondary metabolites or other non-nutritional substances to medicate themselves [16].

The way animals act in the response of a disease or perturbation of homeostasis, can be classified into five types of behavior: (1) “Sick behavior” (lethargy or anorexia); (2) avoidance for prevention of transmission (contaminated food and water); (3) dietary selection of items with a preventative effect (items eaten routinely in small amounts or on a limited basis); (4) intake of a substance for a curative effect (consumption of substances with no nutritional value infrequently and small quantities) and; (5) use of a substance on the skin (fur-rubbing) or habitat construction to control vectors (antibacterial foliage as nest material) [19,20].

Nowadays, it is considered that animal self-medication comprises the levels (3) (Prophylactic-act of using nature’s medicinal resources without any symptoms of infection or before the infections); and (4) (Therapeutic-act of using nature’s medicinal resources only after infection or illness) [16].

Therefore, considering which it is mentioned above, this revision will be focused only on Level (4) since no clear examples to prevent cancer are known.

Level (4) comprises therapeutic behaviors in which only sick individuals are expected to consume pharmacologically active substances from plants. These food items would not be expected to be in the animals’ regular diet, so they lack nutritional value. Moreover, these compounds would probably be more potent than preventive ones, and consequently, would carry a greater risk of negative side effects. This therapeutic behavior requires some level of awareness of wellness and discomfort, and the ability to respond with behaviors that bring about positive change in one’s condition [19,20].

Therefore, the list of the four criteria that must be met to establish self-medication can be summarized in:The substance in question must be deliberately contacted.The substance must be detrimental, in this case, to tumoral cells.The detrimental effect on cancer cells must lead to increased host fitness.The substance must have a detrimental effect on the host in the absence of tumoral cells [18].

The latter is the most important factor in separating self-medication from other behaviors. Criterion 4 can be redefined considering the importance of dose-dependence to state that the substance must be detrimental to unaffected individuals when ingested at the level ingested by infected individuals. 

Another useful distinction is when the substance is consumed since therapeutic self-medication differs from prophylactic self-medication in that consumption occurs after infection [18].

Some anecdotal evidence for self-medication in animals is collected in the works of Huffman [15], Costa-Neto [16], Krief [21] Lozano [20], and Abbott [18] among others.

Emerging evidence suggests ruminants can self-medicate against gastrointestinal parasites by increasing consumption of antiparasitic plant secondary metabolites pharmacologically active, since this selective feeding improves their health and fitness, as it was reviewed by Villalba et al. [22]. One specific example was the work of Lisonbee et al. [23], that showed parasite lambs ate more tannin-containing plants such as the species *Hedysarum coronarium* L. and *Lespedeza cuneata* (Dum.Cours.) G.Don, than lambs without parasites when parasite burdens were high. However, differences became smaller and disappeared toward the end of the study when the amount of parasite decreased [23]. This study was also pointed out that control internal parasite infections through chemotherapy nowadays are failing due to the increased prevalence of parasite resistance to current drugs [23]. MDR is a common problem that sharing both infectious pathologies as well as cancer disease.

Albeit, the most classical example of self-medication behavior is the leaves chewing of *Vernonia amygdalina* Delile (*Asteraceae*) by apes [16]. Particularly, the secondary metabolites present in this plant, such as sesquiterpene lactones (e.g.,: Vernodaline) and steroid glucosides (e.g.,: Vernioside) have shown anthelmintic, antiamoebic, anticancer (inhibition of tumor promotion), immunosuppressive and antibiotic activities [15,16]. Nowadays, it is continued studying the possible therapeutic application and mechanisms (mainly apoptotic) of this extract to the treatment of prostate [24], neuroblastoma [25], and breast cancer, showing in the latter case a synergistic effect in combination with doxorubicin [26]. Besides, other species from the same Genus gather interesting chemical structures with prospective antitumoral activities [27]. 

Another case is the uncommon ingestion of *Albizia grandibracteata* Taub. leaves and bark roots by chimpanzees that showed intestinal problems. Both extracts possess anthelmintic activities revealed by phytochemical studies that also showed the highly significative cytotoxic activity of bark roots extracts justified by the presence of saponosides in high quantities [28]. In the last twenty years, efforts have been continuously made in studying the antitumoral effect of extracts [29,30,31], in elucidating several oleanane-type saponins [32,33,34,35,36,37] within this genus [38], as well as another non-described secondary metabolites [30,39] with potent anticancer activities. It is noteworthy that *Albizia adianthifolia* (Schum.) W.Wight extract showed activity in MDR breast, colorectal carcinoma, glioblastoma, and hepatocellular carcinoma (in MDA-MB-231-BCRP, HCT116 (p53^-^/^-^), U87MG.∆EGFR, HepG2 cancer cell lines respectively) [40].

Furthermore, it has been noticed that some plants are used by animals and humans for the same therapeutic purposes [20]. Some studies compare ethnomedicine/ethnopharmacology and zoopharmacology in the same project [41]. These documented convergencies suggesting that few cases are due to direct observation of sick animal behavior by humans [42]. Perhaps these common behavior patterns are due to phylogenetic closeness (the majority of cases were observed in primates), which justify why men can recognize the physiological activity of the employed plants [13].

For instance, African ethnic groups prepare concoction made from *V. amigdalina* leaves or bark and prescribe it as a treatment for parasitaemias, diarrhea, and stomach upset. This was comparable with the effect of *V. amygdalina* bitter-pith chewing during 20–24 h by chimpanzees which suffered the same symptoms [19].

Another similar study also showed that 13 species of a total of 53 plants present in the diet of woolly spider monkeys (*Brachyteles arachnoides*) population are included in the medicinal repertoire of people living around the same natural area [41].

Recently, it was evidenced that crude extracts of some plants present in Japanese macaque (*Macaca fuscata* yakui) diet possess antiparasitic activity against same of the most important infectious parasites in humans, such as *Trypanosoma brucei* rhodesiense, *Trypanosoma cruzi*, *Leishmania donovani*, and *Plasmodium falciparum*. Besides, many of the studied plants are indigenous Asian species used in traditional medicine for various ailments varying from infections (caused by bacteria, fungi, viruses, helminths as well as endo- and ecto-parasites) to allergy, rheumatism, cancer, and more [13].

Additionally, the work of Dubost et al. [42] recorded numerous confluences between the observation interpreted by mahouts (people in charge of domestic elephants in Asia) as self-medication behavior on the part of the elephants and their medicinal practices. 

The overlapped similarities between the ingestion of plants by primates and their medicinal use by humans provide and could warrant a bio-rational for the search of a bioactive plant in the primate diet with pharmacological and phytochemical value [41]. In 2005, the first work was published that pointed out the pharmacological activities of plant parts eaten by wild chimpanzees (*Pan troglodytes* schweinfurthii)) [43].

Other published works have been focused on the ethnopharmacological study of plants employed for cancer treatment in African countries, showing the use by humans of plant extracts from previously mentioned Genus as medicinal in primates; for instance *Vernonia lasiopus* O.Hoffm., *Albizia coriaria* Oliv., and *A. gummifera* (J.F.Gmel.) C.A.Sm. [44,45]. It should be pointed that some cytotoxic compounds such as Epivernodalol, a sesquiterpene lactone elemanolide-type, present in *Vernonia amydalina* Delile used by chimpanzees, also existed in *Vernonia lasiopus* O.Hoffm. employed by humans [27,46]. Besides, the possible common anticancer compounds among *A. coriaria*, *A. gummifera* (both used by humans), and *A. gradibracteata* used by chimpanzees might be oleanane-type triterpene saponins [28,38,43,44,45].

The most representative case is the swallowing of *Rubia cordifolia* L. leaves by chimpanzees of the Kibale forest that was suspected to possess medicinal value to alleviated abdominal pain caused by intestinal parasites [43,47]. For that reason, the biological activity of its leaves methanol extract on intestinal nematodes (*Strongyloides* spp.) was tested. The results suggested the lack of therapeutic value which could be due to experimental problems that interfered with the test such as the failure of the extraction method, the choice of a non-representative in vitro model that did not simulate the complex interactions resulting from the ingestion if the in vivo activation of secondary compounds would be necessary for antihelmintic action [48], or that its use in primates was as an anti-inflammatory and this activity was not tested. Either way, the genus *Rubia* contains several species used extensively in traditional Chinese, Indian, and Korean medicine, mainly their roots, rhizome extracts, and their phytochemicals have drawn much attention due to their potent bioactivities that include the treatment of cancer, inflammation, infections, rheumatism, and so on [49,50,51,52].

Studies on these plants lead to the isolation of a series of bioactive ingredient including terpenes, cyclopeptides, naphthoquinones, and other constituents [49,52].

Terpenes (such as Rubiatriol) were the earliest phytochemicals isolated but not generally considered as major effective ingredients [47,49]. Compared with terpenes, some cyclic hexapeptides such as Rubicordins A-C showed cytotoxicity against SGC-7901, A549, and HeLa cancer cell lines which could be due to the inhibition of the NF-κβ signaling pathway [53].

By far, naphthoquinones have been extensively studied and gained increasing attention in the last years due to their potent antitumor effects.

Specifically, Mollugin, the major bioactive component isolated from *R. cordifolia* roots acts in NF-κβ transcription factor, a major regulator of the immune response, which is associated with the development and progression of cancer and inflammation. It has been proven that, on the one hand, Mollugin inhibited the expression of NF-κβ reported gen that potentiated TNF-α, a factor that induced apoptosis in a dose-dependent manner, resulting in inhibiting proliferation of HeLa cells [50]. Moreover, it can induce mitochondrial apoptosis and autophagy via inhibition of the PI3K/AKT/mTOR/p70S6K and ERK signaling pathways in glioblastoma cell lines [51]. Also, it was evidenced that Mollugin induces apoptosis in oral squamous cell carcinoma cells through NF-κβ/MAPK/Nrf2/HO1-signalling pathway. HO-1 is overexpressed in various types of cancer and is further induced by radiation and chemotherapy. Regarding the mechanisms of HO-1 induction, several studies have suggested the involvement of ERK/p38MAPK and NF-κB pathways, as well as nuclear factor erythroid 2–related factor 2 (Nrf2) [54]. Mollugin also provoked mitochondrial-derived apoptosis in leukemia Jurkat T cells [52].

Other species of this genus have also attractive naphthohydroquinone compounds such as Rubioncolin C from *Rubia podantha* Diels, which inhibited AKT/mTOR/p70S6K signaling pathway and NF-κβ factor inducing apoptosis and autophagy in HCT116 and HepG2 cell lines [55].

These findings establish *Rubia* naphthoquinones as an attractive possible multitarget therapeutic candidates against human cancer, alone or in combination with other antitumoral agents to overcome drug resistance and achieve better outcomes. However, despite the advanced phytochemical studies on *Rubia* species due to its use in traditional medicine, it remains to be disclosed the therapeutic activity of leaves ingestion on chimpanzees and whether there is a narrow link between animals and local communities medicinal uses.

At last, in the past five years, several papers have been published demonstrating that self-medication is not only present in higher vertebrates, but also it is taxonomically widespread in insects. A summary of cases that convincingly demonstrated self-medication within several different, distantly-related, insect taxa indicated that most representative examples to date are on wooly bear caterpillars (*Grammia rambur*), armyworms (*Spodoptera guenée*), fruit flies (*Drosophila melanogaster*), and monarch butterflies (*Danaus plexippus* Kluk). In these species, all four criteria for demonstrating self-medication have been met [18].

All these observations evidence these types of research projects can give clues to assist in drug discovery with human applications [13]. With the aim of boosting it, the Institute for Ethnobotany and Zoopharmacognosy (IEZ) was founded in 1995 in the Netherlands. Its mission was and is to study and promote the optimal use of wild medicinal and edible plants by humans and animals [56].

As is evidenced, the study of animal self-medication behavior offers a novel and complementary line of multidisciplinary research for targeting plants with bioactive properties for the treatment of resistant pathogens and cancer in humans and their livestock by plant-based medicines [15,20,43].

## 3. Zootherapy

Ancient civilizations such as Egypt and Mesopotamia left historical scripts on which they arose the use of animal-based medicines. Afterward, these medical practices were passed through different regions and appeared in registers that have reached our days. About one-tenth of the remedies mentioned in Dioscorides Materia Medica (100 AD) were animal parts or their products and these types of drugs appear also in the Chinese, Ayurvedic, and Islamic inventories of medicinal substances. The healing of human ailments by using therapeutics based on medicines obtained from animals or, ultimately, derived from them, such as their organs or excretions (venoms), is known as Zootherapy [57,58].

The phenomenon of zootherapy is marked both by a broad geographical distribution and very deep historical origins [58]. Whereas traditional medicines frequently apply whole animals and animal preparations for treatment purposes, modern medicine has recognized the pharmacological value of isolated compounds from animals as leaders for chemical derivatization and further drug development [59].

Despite their importance, studies on the therapeutic use of animals and animal parts have been neglected, when compared to plants [58]. However, in the last three years, some articles have been published compiling the use of animal-based therapies in different places such as India [60,61,62], Latin America [63,64], or Mauritius [65], evidencing the interest that this field arouses nowadays.

In current societies, zootherapy constitutes an important alternative among many other known therapies practiced worldwide [58]. Thus, to combat the epidemics of deadly diseases such as cancer, there is an essential need to identify efficacious agents with a novel mode of actions from unless exploited natural sources [66].

As a result, several studies on terrestrial mammals antitumoral by-products have been carried out.

For example, one report has described that four bovines (*Bos primigenius* taurus) meat-derived peptides (GFHI, DFHING, FHG, GLSDGEWQ) exhibit the most cytotoxic effect on MCF-7 breast human cancer cell line and decrease the viability of stomach adenocarcinoma cell lines (AGS) [67]. Also, a recent work mentioned some antineoplastic effect in S180, 4T1, and HepG2 cell lines of dried gallstones of domesticated cows (*Bos Taurus domesticus* Gmalin), also denominated as *Calculus bovis* [68]. From other livestock animals like goats (*Capra aegagrus* hircus) spleens or livers were isolated the bioactive peptide-3 (ACPB-3), that exhibited anticancer activity against BCG-823, GCSC, and HCT-116 cell lines in vitro and in vivo [69]. Moreover, numerous studies have reported the anticancer effects of milk derived-peptides, such as casein, transferrin, and lactoferricin [69].

Furthermore, by-products derived from birds such as the blood ingestion derived from American Black Vulture (*Coragyps atratus*) appear to show anticancer activity against hematopoietic system cancer cell lines [70].

Some reptiles products are also being evaluated for this purpose. Siamese crocodiles (*Crocodylus siamensis*) are one of the most studied species since their bile acid extract inhibited the proliferation of SK-ChA-1, Mz-ChA-1, QBC939, and SMM 7721 cell lines in a dose-dependent manner and besides, enhance the sensitivity of drug uptake by human cholangiocarcinoma MDR cell line (QBC939/5-FU). In addition, their white blood cell extracts exhibited anti-angiogenic properties in HeLa cell lines [71]. It was shown that ESC-3 was the active component in crocodile bile (as it was studied in SK-ChA-1, Mz-ChA-1, QBC939 cell lines), that induced apoptosis in the former mentioned cell line [72]. Also, from whole animal aqueous extracts of *Gekko swinhonis* Güenter was isolated the Gekko Sulfated Glycopeptide or Gepsine. Gepsine exhibited anti-proliferative activity in Bel-702 and HT-29 cell lines; besides, it showed anti-angiogenic effects in human lymphatic endothelial cells (hLECs) due to disruption in bFGF function, which is a growth factor responsible for angiogenesis [73,74]. Moreover, the enzymatic hydrolysates of the three-striped box turtle (*Cuora trifasciata*) inhibit HepG2 and MCF-7 cells [75].

Mollusks and crustaceans also appear to be interesting in this area. Data demonstrated that the aqueous extract of garden snail (*Helix aspersa*) has antitumor activity against breast cancer cell line H5578T inducing necrosis; besides, it is a potent stimulator for TNF-α and a good inhibitor for NF-κβ, PTEN and p53 factors that regulate tumor development [76]. Table 1 compiled the previously mentioned examples of terrestrial animal anticancer by-products.

In the past two decades, many efforts have been put in discovering new compounds from marine species of tunicates, ascidians, sponges among other organisms [69,77]. Marine pharmacology is the new discipline that explores the marine environment searching for potential pharmaceuticals [77].

Several reports indicated the potential antiproliferative activity of several peptide fractions derived from different fish species, as it was compiled in the work of Wang et al. [69]. Furthermore, the lipid fraction of white shrimp (*Litopenaeus setiferus*) contains compounds that have been found to reduce the proliferation of a B-cell lymphoma cell line. Besides, shrimp anti-lipopolysaccharide factor (SALF) from black tiger shrimp (*Penaeus monodon*), enhancing the anticancer activity of cisplatin in vitro and inhibiting HeLa cell growth in nude mice [69].

However, the most known examples in this area are represented by the Trabectedin (Yondelis^®^, Ecteinascidin 743) from the tunicate *Ecteinascidia turbinata* which destabilizes and inhibits tumor cells DNA; the Aplidine (isolated from another tunicate, *Aplidium albicans*) that inhibits cell division [59,77] and it is been studied as a treatment for Multiple Myeloma in conjugation with Dexamethasone [9]; or the Conotoxin (k-PVIIA) from *Conus purpurascens* which increased hERG expression, a non-classical drug target for anticancer drugs [59].

Amidst the compounds semi-/synthetized from marine drugs (some of them are currently under clinical trials evaluation) stand out the Eribulin, a derivate from Helichondrin B, a sponge metabolite isolated in Japan [78]; the Lurbinectedin (which derived from Trabectedin) and the Plocabulin isolated from the sponge *Lithoplocamic lithistoides* [9]. 

Despite the thriving developments in this field in the past years, comparatively few investigations focused on the collateral sensitivity of these natural products. Cell lines resistant to diverse antitumoral common drugs such as cisplatin, vinblastine, or paclitaxel, revealed collateral sensitivity to marine substances such as aragusterol A, cytochalasin B, ningalin N3 and N5, dolastatin-10 and the hemiasterlin derivative HTI-286 [8]. Some reviews also highlighted several marine natural products with a reversal effect on multidrug resistance in cancer mediated by drug transporter proteins, such as the works of Abraham et al. [79] and Lopez and Martínez-Luis [80]. 

Finally, one of the appealing strategies in oncology within this area consists of the study of venoms and their isolated compounds. Venoms are the poisonous secretions that are produced by specialized glands associated with teeth, stings, and spines of the respective animal. The composition of venom differs from animal to animal, most of the venoms being a heterogeneous mixture of inorganic salts, low molecular weight organic molecules, peptides, and enzymes [66].

In recent times, several studies have provided evidence that despite the toxicity of venoms, their biotoxins can be exploited as a source from which novel anticancer agents with innovative mechanisms of action may be developed [66,81] (Table 2). 

Among the most studied venoms are the ones produced by amphibians, reptiles, and insects.

Amphibians venoms are rich in a wide range of chemical compounds including steroids (bufadienolides such as marinobufagin, telocinobufagine, bufalin), amines (like bufotenine), proteins, peptides and different classes of alkaloids [82]. A recent study showed that the venoms of *Rhinella schneideri* and *R. marina* exhibit a strong tumor growth inhibition toward drug-sensitive CCFR-CEM and P-glycoprotein overexpressing CEM/ADR5000 leukemia cells. Moreover, *Venenum Bufonis* (VB), a product of the secretions of *Bufo gargarizans* Cantor and *B. melanostictus* Schneider, which has been long used as Traditional Chinese Medicine exhibit antitumor, antinociceptive and anti-inflammatory activities in different types of cancer [83]. It appears that these effects are mainly due to bufadienolides [82,83]. Within these metabolites, the Arenobufagin (ArBu) isolated from VB, targets IKKβ/ NF-κβ signal cascade resulting in inhibition of Epithelial-to Mesenchymal Transition and suppressed migration and invasion of lung cancer cells [84]. Likewise, some products from the frog *Physalaemus nattereri* act against invasive cells [71].

Another important source of new compounds is snake venoms composed by metalloproteases (MPP), disintegrins (such as Contortrostatin, tzascanin, DisBa-01), L-Amino Acid Oxidases, C-type lecitins (BJcuL, Lebecin, Daboialectin), polypeptides (Crotamine, Cathelicidin-BF, Cardiotoxin III, Ancrod), phospholipases A2, Acetyl Cholinesterases and Serine proteases (Batroxobin). Moreover, snake venoms composition vary depending on the family and can be grouped according to their pathophysiological activities as follows: (i) *Elapidae* venoms (example: *Naja naja*) which cause irreversible alterations on the cell, destroying it; (ii) *Crotalidae* venoms (*Crotalus* spp., *Bothrops* spp.) which cause loss of the cell process viability; and (iii) *Viperidae* venoms (*Macrovipera* spp.), which cause alterations of cell aggregation [85,86].

Among the insects, stands out the medicinal use of scorpions, beetles, spiders, bees, wasps, and caterpillars.

The Yellow Israeli Scorpion (*Leiurus quinquestriatus* quinquestriatus) venom contains a peptide molecule, the chlorotoxin (CTX), that blocks small conductance Cl^−^ channels, diminished annexin-2 expression, and inhibits MPP-2 in malignant cells. CTX has shown specificity to bind to neuro-ectodermal descent tumors like glio-, neuro-, and medulloblastomas, melanomas, small cell lung carcinomas among others, and no activity in normal brain cells. Nowadays, it has been proved that the combination of CTX with drugs such as methotrexate or cisplatin increases the cytotoxicity toward cancer cells when compared to the effect of the drug alone [59,87]. Oher scorpion venoms and their possible mechanisms of action are summarized in the paper of Ding et al. [88].

The blister beetles *Epicauta hirticornis* and different species of *Mylabris* (*M. cichorii*, *M. phalerata*) posses Cantharidin, a terpenoid able to induce intrinsic apoptosis, necrosis and autophagy cell death in Ehrlich Ascites Carcinoma (EAC) cells [89,90]. Recently, thirteen new cantharidin derivatives were isolated from *M. phalerata* [90].

As in previous cases, spider venoms are a complex mixture being of interest in oncotherapy by the peptides they content such as Brachynin, Lycosin-I, and Gomesin presented in the species *Brachypelma albopilosum*, *Lycosa carolinensis* and *Acanthoscurria gomesiana* respectively [91].

**Table 2 biomolecules-10-00817-t002:** Summary of mentioned examples of animal venoms. Abbreviation: Ref, Reference.

Group of Animals	Order	Family	Representative Species	Bioactive Compounds Found in Venoms	Ref.
Amphibians	*Anura*	*Bufonidae*	*Rhinella schneideri, R. marina*	Different types of steroids (bufadienolides), amines, proteins, peptides, alkaloids	[71,82,83,84]
			*Bufo gargizans Cantor, B. melasnosticus*		
		*Leptodactylidae*	*Physalaemus nattereri*		
Snakes	*Squamata*	*Elapidae*	*Naja naja* among others	MPP, disintegrins, L-amino acid oxidase, C-type lecitins, polypeptides, phospholipase A2, acetyl cholinesterases, serine proteases	[85,86]
		*Crotalidae*	*Crotalus spp.*, *Bothrops spp.*		
		*Viperidae*	*Macrovipera spp.*		
Mammals	*Eulipotyphla*	*Soricidae*	*Blarina brevicauda*	Soridin	[92,93]
Marine (Jellyfish)	*Semaeostomeae*	*Pelagiidae*	*Chrysaora quinquecirrha*	SNV peptide	[59]
Insects	*Scorpiones*	*Buthidae*	*Leiurus quinquestriatus quinquestriatus*	Chlorotoxin	[59,87]
	*Coleoptera*	*Meloidae*	*Epicauta hirticornis*, *Mylabris cichorii*, *M. phalerata*	Cantharidin	[89,90]
	*Araneae*	*Theraphosidae*	*Brachypelma albopilosum*	Brachynin	
			*Acanthoscurria gomesiana*	Gomesin	
		*Lycosidae*	*Lycosa carolinensis*	Lycosin-I	
	*Hymenoptera*	*Apidae*	*Apis mellifera*	Melittin	[94,95,96]
		*Vespidae*	*Polybia paulista*	Mastoparan	
	*Lepidoptera*	*Saturniidae*	*Hyalophora cecropia*	Cecropin	[96]

Hymenoptera venoms, those which come from bees (*Apis mellifera*) and wasps (*Polybia paulista*) contain Melittin and Mastoparan as the main important compounds respectively, because of their evidenced anticancer properties as it has been reviewed in some works [94,95]. Moreover, both compounds showed activity in MDR cells and improve the therapeutic activity in combination with drugs [94,95]. A novel Mastoparan peptide, Polybia-MPI from *Polybia paulista* wasp venom can induce cell death targeting non-polar lipid cell membranes [96].

Lastly, Cecropin, a type of peptides derived from the hemolymph of the giant silk moth (*Hyalophora cecropia*) demonstrated activity against mammalian leukemia, lymphoma, colon carcinoma, small cell lung, and gastric cancer cells [96].

In other animals such as mammals, the Soridin, a toxic peptide that selectively inhibits TRPV6 channels, has been isolated from shrew (*Blarina brevicauda*). TRPV family channels have been implicated in tumor development and progression in many carcinomas of ovary, prostate, thyroid, colon, and breast [92]. Finally, studies found that sea nettle nematocyst venom (SNV) peptide from *Chrysaora quinquecirrha* possessed a significant antitumor effect on EAC tumors [93].

## 4. Future Perspectives

Cancer is a multifactorial and multitarget disease treated, at present, mainly employing chemotherapy, radiotherapy, and immunotherapy. However, these therapies carry the appearance of detrimental effects mainly due to MDR. Understanding how the switch between sensitive and resistant states is activated and what factors control it would be key to designing new effective treatment strategies for MDR cancers [3]. 

Natural products have long served as sources of therapeutic drugs and as a point of inspiration for researchers for the development and design of new drug candidates [63,97]. In this sense, Ethnopharmacology-based approaches always have been a promising strategy in drug discovery. The present work is a starting point that summarizes, explains and exemplifies how Zoopharmacognosy and Zootherapy can meet this goal. Some of the most interesting mentioned compounds from Zoopharmacognosy are triterpene saponins present in genus *Vernonia* and *Albizia,* and naphthoquinones such as Mollugin and Rubioncolin C. Also, from Zootherapy origin, several steroids from amphibians and peptides from different mammals, reptiles and insects stand out. An excellent work that is an example of insect potential pointed out that there are at least 16 times as many insect species as there are plant species, yet plant chemistry has been studied 7000 times as much as insect chemistry when comparing the amount of research per species. Nonetheless, the vast biodiversity which exists in the arthropod world, compared to all other organisms on earth, certainly suggested that arthropods should be given a more serious look [63].

Because of their unique structural features, natural products remained promising to identify active compounds for tough targets, which were difficult to direct with classical small molecules [97].

In recent years, innovative in silico strategies have contributed valuably to these research processes [97]. A computational field known as “virtual screening” (VS) has emerged in the past decades to aid experimental drug discovery studies by statistically estimating unknown bio-interactions between compounds and biological targets [98]. VS can be mainly divided into ligand-based (if it uses the molecular properties of compounds to model interactions with targets) and ligand-based (if it employs the 3D structure of targets and compounds to model the interactions) [98]. It involves the computational filtering of a large body of molecules (e.g., those comprising a database) to identify those that have a high probability of activity in the biological test system of interest [99]. This is done through machine learning techniques that generate predictive models [98]. The concept of molecular similarity lies at the heart of such methods since they can discriminate between active and inactive test-set molecules if there are some structural commonalities (in terms of the descriptors available) between the training-set actives and/or structural dissimilarity between the training-set actives and inactives [99]. The application of Machine learning similarity-based methods relies on the assumption that biologically, topologically and chemically similar compounds have similar functions and bioactivities and, therefore, they have similar targets [98]. Recently, it was presented a resource named Chemical Checker (CC) which provides processed, harmonized, and integrated bioactivity data (from chemical properties to their clinical outcomes) on ~800,000 small molecules. This tool is based on the so-called “similarity principle” previously mentioned (similar compounds not only show similar chemical properties but also show similar biological behavior. Molecules with similar cellular sensitivity profiles, or causing similar side effects, often share the same mechanism of action, even when their chemical structures appear unrelated). Thus, CC can aid drug discovery tasks, including target identification and library characterization [100].

Aside from this, there are some VS studies with natural substances. In 2018, it was performed ligand-based and structure-based virtual screens of 1306 sesquiterpene lactones of *Asteraceae* (e.g., *Vernonia* spp.) obtained from an in-house database (SistematX database). Potential antichagasic of these secondary metabolites for the three parasitic forms and some structural features, such as skeleton type and the presence of epoxide moiety, were determined from random forest models of *T. cruzi*. Also, a structure-based virtual screen using PDB structures of eight *T. cruzi* proteins for the entire sesquiterpene lactones set allowed the selection of 13 potential inhibitors of these enzymes. Finally, an approach combining structure-based and ligand-based virtual screening enabled the identification of promising single and multitarget antichagasic sesquiterpene lactones [101].

Other studies employed machine learning methods to determine the biomarkers with anti-inflammatory potential (dual inhibition of COX and LOX) in extracts of 57 *Asteraceae* species. Moreover, models to detect biomarkers in new extracts based solely on their metabolomic data, and with no prior knowledge of biological data, were also established. This combined strategy that uses metabolomics, in vitro bioactivity, decision trees, and artificial neural networks had never before been explored; therefore, the time-and money-consuming steps usually required for compound isolation, identification, and anti-inflammatory evaluation were avoided. This strategy could be useful in studies that seek to find biomarkers with a certain determined property or characteristic of complex samples (such as the extracts) and also to reveal active secondary metabolites [102].

In this regard, if it is known which targets are specifically involved in a certain type of cancer, and all the information until this moment of the compounds discovered using Zoopharmacology would be gathered in databases, it will be possible to interrelate them by inverse docking tools (such as Selnergy), in a process called as Reverse Pharmacognosy. Reverse pharmacognosy allows us to identify which molecule(s) from an organism is (are) responsible for the biological activity and the biological pathway(s) involved. This approach provides evidence of the therapeutic properties of products used in traditional medicine. Moreover, new activities can be discovered for the ligands, which would allow the molecule reposition, which consists of finding applications unknown until now for the plant constituents studied, which would be subsequently validated by in vitro and in vivo tests. Also, by ascribing the new properties studied to a molecule, all the standardized extracts of the organisms that contain it at the same concentration would be repositioned, as well as its adverse effects. Finally, the activity of the ligand could be optimized based on the characteristics it presents, looking for derivatives with similar properties in the same raw material or others, that are more potent, less toxic, with greater accessibility to be exploited or with a better pharmacological profile of the selected molecule, which, in short, would allow obtaining active extracts and, with it, new patents related to them [103].

In terms of the methodological approach used in modeling the pairwise relationships, a highly studied topic is the development of network or graph analysis-based drug target interaction (DTI) prediction methods. In these methods, compounds and targets are represented as nodes on a graph, where the edges connecting these nodes indicate interactions. Modeled this way, the estimation of unknown DTIs becomes a link prediction task. A trend in DTI prediction that we expect to become more popular soon is integrating large-scale omic at the input level, to increase both the quality and the coverage of the predictions [98]. This concerns examples of network pharmacology.

A study investigated the antihepatoma components and mechanism of action of *Rubia cordifolia* L. Its chemical components were evaluated and screened comprehensively and efficiently by drug-likeness and pharmacokinetic characteristics. The target sites of multiple active components were predicted according to the reverse pharmacophore matching model and antihepatoma-related gene targets of *Rubia cordifolia* L. were screened through comparison with the databases. The functions of target genes and related pathways were analyzed and screened, and the component/target/pathways network of anti-liver cancer effect was constructed. The pathway enrichment results indicated that *Rubia cordifolia* L. may play an antihepatoma role by inhibiting the development of hepatitis B or acting on the key targets of the MAPK and PI3K/AKT signaling pathway, such as MAPK1, HRAS, and AKT1. The network was used to evaluate the antihepatoma activity of various components. The results of network pharmacology were further verified by molecular docking of selected active components and the key targets. The present findings guided further pharmacological studies on this plant [104].

One of the latest approaches consisted of the lipidomics-proteomics study of the mechanism of action of ArBu on liver cancer and hepatoma cells to clarify its mechanism by bioinformatic methods. ArBu regulated lipid homeostasis which might be closely related to the mechanism of hepatocellular carcinoma. The lipidomics research revealed that lipid concentrations were regulated after the intervention of ArBu. Glycerophospholipid metabolism was the major and commonly affected pathway. ArBu was the potential inhibitor of JAK-STAT3 564 signaling pathway, which could induce apoptosis and promote autophagy of HepG2 cells. Additionally, ArBu could cut off the supplies of multiple amino acids in HepG2 cells and affect the progression of liver cancer as a potential arginine deiminase agonist. In summary, these results provided a rationale for a comprehensive micro-environment regulation on the anti-hepatoma effects of ArBu [105].

Finally, bioprospecting practices have implications for medicine, environment, economy, public health, and culture [63]. The potential benefits of natural products-based medicines have led to the unscientific exploitation of these resources, a phenomenon that is being observed globally. This decline in biodiversity and natural loss due to species extinction is largely the result of the rise in the global population, rapid and sometimes unplanned industrialization, indiscriminate deforestation, overexploitation of natural resources, illegal trade, pollution, and finally global climate change [63,106]. Some particular aspects should be considered in Zootherapy practices such as animal products are, in most cases by-products from animals hunted for other purposes; their use is common for medicinal, religious beliefs and socio-cultural practices in both rural and urban areas of Asia, Africa or Latin America; it exists the possibility of transmitting infections or ailments from animal traditional preparations to the patient and they remain virtually unstudied and undocumented that entails their potential knowledge disappearance [63]. Therefore, it is of utmost importance that biodiversity is preserved, to provide future structural diversity and lead compounds for the sustainable development of human civilization. This becomes even more important for developing nations, where well-planned bioprospecting coupled with non-destructive commercialization could help in the conservation of biodiversity, ultimately benefiting mankind in the long run [106].

## 5. Conclusions

Cancer is a multifactorial and multitarget disease treated, at present, mainly using chemotherapy, radiotherapy, and immunotherapy. However, these therapies carry the appearance of detrimental effects mainly due to MDR. Understanding how the switch between sensitive and resistant states is activated and what factors control it would be key to designing new effective treatment strategies for MDR cancers. Ethnopharmacology-based approaches always have been a promising strategy to find new drugs. The present work is a starting point that summarizes, explains and exemplifies how Zoopharmacognosy and Zootherapy can meet this goal. Currently, Zoopharmacognostic studies have to improve in many ways: compilation of more case descriptions, evaluation of the potential for self-medication in more taxa, and determination if there are any general evolutionary or ecological predictors of the behavior; increasing phytochemical and bio-guided activity studies of a plant used by animals and searching for correlations between the use of medicinal plants of humans and animals. Great expectations have been put in secondary metabolites such as saponins and naphthoquinones unveiling in these studies as cancer lead agents. The on-going tendency in Zootherapy is focused mainly on exploring marine drugs and in keeping records of the traditional medicinal uses of animal products. However, neither a demonstration of the clinical efficacy of the popularly used remedies nor an evaluation of the sanitary implications of the prescription of animal products for the treatment of diseases was studied. In both cases, the number of studies must be increased, the techniques improved and protection measures must be promoted for these species, using research methods that cause the least possible impact and damage to the species, local community, and environment. Henceforth, intensive cross-disciplinary research approaches including chemistry, pharmacology, molecular biology, etc. that encompassing the conservation and sustainability of the species, could realize the Zoopharmacology one of the mainstreams in Oncotherapy.

## Figures and Tables

**Figure 1 biomolecules-10-00817-f001:**
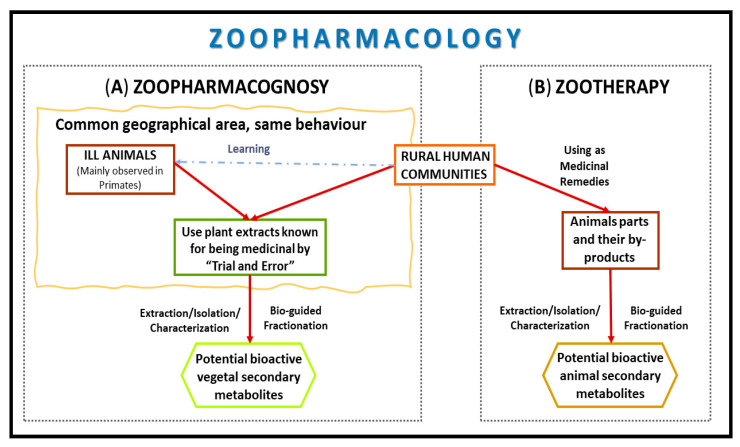
Zoopharmacology strategy involves drug development routes of (**A**) Zoopharmacognosy and (**B**) Zootherapy for anticancer drugs.

**Figure 2 biomolecules-10-00817-f002:**
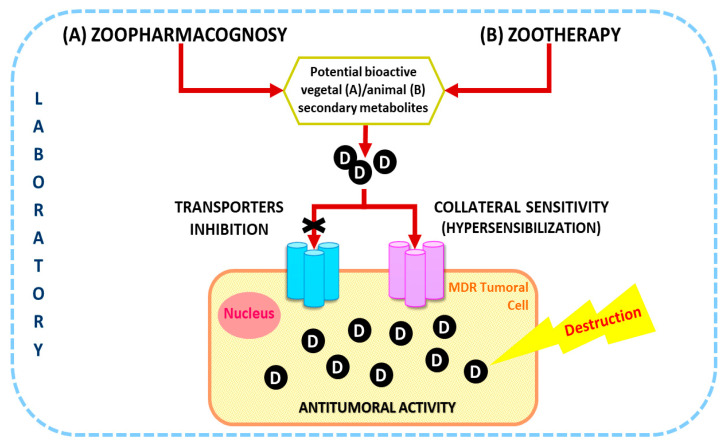
Scheme explains how the bioactive compounds provided by Zoopharmacognosy and Zootherapy drug development strategies can overcome MDR. The strategies involve the inhibition of transporters that expel the drugs and the hypersensitization of cancer-resistant cells. Abbreviation: D, Drug; MDR, MultiDrug Resistance.

**Table 1 biomolecules-10-00817-t001:** Summary of mentioned examples of terrestrial animal anticancer by-products. Abbreviation: Ref, Reference.

Animal Specie	Product	Molecule(s)	Type of Cancer	Studied Cell Line	Ref.
*Bos primigenius* taurus	Meat peptides	GFHI, DFHING, FHG, GLSDGEWQ	Breast	MCF-7	[67]
			Stomach adenocarcinoma	AGS	
*Bos Taurus* domesticus *Gmalin*	Dried gallstones	Not specified	Sarcoma	S180	[68]
			Breast	4T1	
			Hepatoma	HepG2	
*Capra aegagrus* hircus	Spleen, liver extracts	ACPB-3	Gastric	BCG-823	[69]
				GCSC	
			Colorectal	HCT-116	
*Coragyps atratus*	Blood extracts	-	Hematopoietic system	-	[70]
*Crocodylus siamensis*	Bile acid extracts	ESC-3	Cholangiocarcinoma	SK-ChA-1	[72]
				Mz-ChA-1	
				QBC939	
		Bile components	Human papillomavirus-related endocervical adenocarcinoma	SMM 7721	[71]
			Cholangiocarcinoma resistant to 5-Fluorouracil	QBC939/5-FU	
	White blood cell extract	-	Cervical Cancer (Anti-angiogenic properties)	HeLa	
*Gekko swinhonis* Güenter	Whole animal aqueous extracts	Gepsine	Hepatocarcinoma	Bel-702	[73,74]
			Colon carcinoma	HT-29	
			Anti-angiogenic properties	hLECs	
*Cuora trifasciata*	Enzymatic hydrolysates extracts	Fraction T1 (peptides T1-1 RGVKGPR, T1-2 KLGPKGPR), Fraction T2 SSPGPPVH	Liver	HepG2	[75]
			Breast	MCF-7	
*Helix aspersa*	Snails aqueous extracts	Not specified	Breast	H5578T	[76]

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
