# Peer review of "Zoopharmacology: A Way to Discover New Cancer Treatments"

_biomolecules, 2020, doi:10.3390/biom10060817_

Round 1
Reviewer 1 Report
I suggest the authors to explain and discuss in this manuscript that there are studies that compare the two areas in the same project: zoopharmacognosy and ethnopharmacology / ethnomedicine, as well as the reference already mentioned by the authors in this manuscript (Petroni et al., 2017). And also another very important reference, which apparently is not mentioned in this manuscript (Krief et al., Ethnomedicinal and bioactive properties of plants ingested by wild chimpanzees in Uganda, 2005).
I suggest bringing a discussion about these studies at some point in the manuscript, since it can contribute even more in the selection of potential bioactive substances to be investigated by chemistry and pharmacology. But it is just a suggestion to further improve this manuscript, which is already very beautiful.
Author Response
We thank and appreciate the reviewers’ comments, which have helped us to improve the manuscript. We have carefully considered the suggestions, addressing and incorporating them in the manuscript as detailed below. The modifications have been marked in yellow and the mentioned lines are referring to the new submitted version of the manuscript.
Comment 1: I suggest the authors to explain and discuss in this manuscript that there are studies that compare the two areas in the same project: zoopharmacognosy and ethnopharmacology / ethnomedicine, as well as the reference already mentioned by the authors in this manuscript (Petroni et al., 2017).
Authors: Thank you for the comment. The suggested comment was included between lines (210-211) as “(… same therapeutic purposes [20].) In fact, some studies compare ethnomedicine/ethnopharmacology and zoopharmacology in the same project [41]. (These documented convergencies suggesting that few cases are due to direct observation of sick animal behaviour by humans [42]…)”.
Comment 2: And also another very important reference, which apparently is not mentioned in this manuscript (Krief et al., Ethnomedicinal and bioactive properties of plants ingested by wild chimpanzees in Uganda, 2005).
Authors: Thank you for the reference, it is indeedn important reference. The reference has been included [43] (lines 234-236; 243-245).
Comment 3: I suggest bringing a discussion about these studies at some point in the manuscript since it can contribute even more in the selection of potential bioactive substances to be investigated by chemistry and pharmacology. But it is just a suggestion to further improve this manuscript, which is already very beautiful.
Authors: Thank you for the suggestion. Considering these indications, it has been carried out in-depth research to improve the whole Zoopharmacognosy section (lines 131-303) and now the authors considered that it has been answered and included the proposed improvements.
Additional authors notes for the reviewer:
- The sentence “The overlapped similarities between the ingestion of plants by primates and their medicinal use by humans provides and could warrant a bio-rational for the search of bioactive plant in the primate diet with pharmacological and phytochemical value [41].” place changes from lines 166-168 to 232-234.
- Table 1 format (lines 360-363) has been re-adapted to the new version of the manuscript.
- It was created a new section, Section 4 title change from 4. Conclusions to 4. Future perspectives.
- Conclusions is now section 5.

Reviewer 2 Report
This manuscript reviews information about how animal self-medication and animals parts/products can be used in the development of novel cancer therapies. There are numerous minor grammatical and spelling error throughout, so it would be helpful to engage the services of a copyeditor when revising the manuscript. Apart from this I found the overall content to be interesting and potentially useful. What I was mainly missing was some sort of summary of general patterns and trends. What types of compounds seem to have particularly high potential and why? How can we use this knowledge to look for similar compounds in a targeted way? Should we be particularly interested in compounds that come from animals which rarely develop cancer in captivity, for example?
I was also a bit confused by the discussion of multidrug resistant cancer types. Obviously this is a problem, but surely there is no horizontal transmission of drug resistance similar to antibiotic resistance, and that resistance arises independently within individuals during the course of treatment. If so, why is MDR a particular problem now compared to before? Is it because treatments have become more effective so that patients live longer, which gives a larger window of opportunity to develop resistance? Or it is a byproduct of some other factor, e.g. off-target effects of medicines that are used routinely prior to cancer diagnosis? I think this issue merits a bit more discussion, especially how it relates to the usefulness of animal-based/inspired treatments.
Minor comments
Figure 1: It might be useful to include a broken line between humans and animals in the left side of the figure, to indicate how humans might learn to identify active substances by observing local animals species.
Lines 78-80: I'm a bit skeptical that this is the first time this has proposed, unless you specifically mean in the context of MDR cancer.
Lines 116-118: In Abbott 2014 (https://onlinelibrary.wiley.com/doi/full/10.1111/een.12110) there is a discussion of the importance of identifying a negative effect of consumption of the substance in the absence of illness, order to determine that it is truly an example of self-medication. It might be worth bringing this issue up in more detail here. Sometimes animals consume specific substances because they need more of a given type of nutrient when ill, but the nutrient does not have a curative effect in itself. Particularly bioactive compounds with a negative effect on cancer are likely to be dangerous to consume in large amounts when unaffected, so this could be an important property to consider when trying to identify potentially useful compounds. (This reference might also be appropriate to cite on lines 94 and 121.)
Lines 253-255: Sponges seem to have a rather unique immune system that doesn't build on specialized immune cells, but rather excretion of biologically active compounds throughout the organism (i.e. a type of humoral immunity). Such compounds might be more likely to have broad-scope effects than those associated with cell-mediated immunity, such as is common in mammals. Insects also use humoral immunity rather than cell-mediated immunity and antibodies. Perhaps the species with an immune system which is least similar to humans could be of most interest? This could be worth mentioning.
Author Response
Comment 1: This manuscript reviews information about how animal self-medication and animals parts/products can be used in the development of novel cancer therapies. There are numerous minor grammatical and spelling error throughout, so it would be helpful to engage the services of a copyeditor when revising the manuscript.
Authors: Thank you for the correction, the English language has been corrected.
Comment 2: Apart from this I found the overall content to be interesting and potentially useful.
Authors: Thank you for the reviewer attention that help us to improve the manuscript. According to these indications, it was written a section of perspectives (lines 459-574) that mentioned the most interesting compounds and methods to look for similar compounds, and in the conclusion was summarised the general patterns and trends (lines 459-599).
Comment 3: What I was mainly missing was some sort of summary of general patterns and trends.
Authors: According to this indication, it was written a section of perspectives (lines 459-574) and in the conclusion was summarised a summary of general patterns and trends (lines 576-599): Cancer is a multifactorial and multitarget disease treated, at present, mainly by means of chemotherapy, radiotherapy and immunotherapy. However, these therapies carry the appearance of detrimental effects mainly due to MDR. Understanding how the switch between sensitive and resistant states is activated and what factors control it would be key to designing new effective treatment strategies for MDR cancers. Ethnopharmacology-based approaches always have been a promising strategy to find new drugs. The present work is a starting point that summarize, explain and exemplify how Zoopharmacognosy and Zootherapy can meet this goal. Currently, Zoopharmacognostic studies have to improve in many ways: compilation of more case descriptions, evaluation of the potential for self-medication in more taxa, and determination if there are any general evolutionary or ecological predictors of the behaviour; increasing phytochemical and bio-guided activity studies of plant used by animals and searching for correlations between the use of medicinal plants of humans and animals. Great expectations have been put in secondary metabolites such as saponins and naphthoquinones unveiling in these studies as cancer lead agents. The on-going tendency in Zootherapy is focused mainly on exploring marine drugs and in keeping records of the traditional medicinal uses of animal products. However, neither a demonstration of the clinical efficacy of the popularly used remedies nor an evaluation of the sanitary implications of the prescription of animal products for the treatment of diseases was studied. In both cases, the number of studies must be increased, the techniques improved and protection measures must be promoted for these species, using research methods that cause the least possible impact and damage to the species, local community and environment. Henceforth, intensive cross-disciplinary research approaches including chemistry, pharmacology, molecular biology, etc. that encompassing the conservation and sustainability of the species, could realize the Zoopharmacology one of the mainstreams in Oncotherapy.
Comment 4: What types of compounds seem to have particularly high potential and why?
Authors: As we mention in several parts of the manuscript, there are a huge diversity of possible substances that can be discovered from Zoopharmacology, either of plant origin by means of Zoopharmacognosy, or by animals (Zootherapy).
These types of secondary metabolites have showed some biological properties as anti-inflammatory (i.e.: phenyl propanoids, flavonoids, etc.), antimicrobials, antivirals, antiparasitic (ej: saponins, lactonic sesquiterpenes…) that share mechanisms of action that can participate or be involved in the control of various pathologies at the same time. Moreover, they can inhibit drug transporters to avoid drug extrusion.
Some of the most interesting mentioned compounds from Zoopharmacognosy are triterpene saponins present in Genus Vernonia and Albizia, naphtoquinones such as Mollugin (it acts in NF-κβ transcription factor, a major regulator of the immune response, which is associated with the development and progression of cancer and inflammation) and Rubioncolin C (which inhibited AKT/mTOR/p70S6K signalling pathway and NF-κβ factor inducing apoptosis and autophagy in HCT116 and HepG2 cell lines). Also, from Zootherapy origin, stand out steroids from amphibians and peptides from different mammals, reptiles and insects.
Comment 5: How can we use this knowledge to look for similar compounds in a targeted way?
Authors: It is possible to look for similar compounds in a targeted way by virtual screening using machine learning techniques. The concept of molecular similarity lies at the heart of such methods, since they can discriminate between active and inactive test-set molecules if there are some structural commonalities (in terms of the descriptors available) between the training-set actives and/or structural dissimilarity between the training-set actives and inactives. According to molecular similarity principle, similar compounds not only show similar chemical properties, but also show similar biological behavior. Molecules with similar cellular sensitivity profiles, or causing similar side effects, often share the same mechanism of action, even when their chemical structures appear unrelated). (see: Evaluation of machine-learning methods for ligand-based virtual screening; https://link.springer.com/article/10.1007%2Fs10822-006-9096-5. Recent applications of deep learning and machine intelligence on in silico drug discovery: methods, tools and databases; https://academic.oup.com/bib/article/20/5/1878/5062947.). An interesting resource based on molecular similarity approach is Chemical Checker : Extending the small-molecule similarity principle to all levels of biology with the Chemical Checker, https://doi.org/10.1038/s41587-020-0502-7
Some examples of these in silico methods with natural products are:
- In silico studies designed to select sesquiterpene lactones with potential antichagasic activity from an in-house Asteraceae database, https://chemistry-europe.onlinelibrary.wiley.com/doi/full/10.1002/cmdc.201700743
- Prediction of Anti-inflammatory Plants and Discovery of Their Biomarkers by Machine Learning Algorithms and Metabolomic Studies, https://www.thieme-connect.com/products/ejournals/abstract/10.1055/s-0034-1396206
Besides, it is possible to predict the therapeutic response by means of omics network studies (metabolomics, proteomics, fluxomics, etc.). Some examples with natural products have been included in the manuscript:
- Network pharmacology-based research on the active component and mechanism of the antihepatoma effect of Rubia cordifolia: https://onlinelibrary.wiley.com/doi/abs/10.1002/jcb.28513
- The lipid homeostasis regulation study of arenobufagin in zebrafish HepG2 xenograft model and HepG2 cells using integrated lipidomics-proteomics approach: https://www.sciencedirect.com/science/article/abs/pii/S0378874120300477
These are some approaches to consider since the study natural products using bioinformatics has suffered currently a great expansion in the last years.
(Supplementary information: Approaching Complex Diseases: Network-Based Pharmacology and Systems Approach in Bio-Medicine, https://link.springer.com/book/10.1007%2F978-3-030-32857-3)
In light of these guidelines, the subject was developed in the section 4. Future Perspectives.
Comment 6: Should we be particularly interested in compounds that come from animals which rarely develop cancer in captivity, for example?
Authors: It appear to be obvious that captivity conditions can affect animals in some ways more or less stressful. However, we do not know how these types of conditions can affect their ability to produce molecules that help repair or inhibit cancer processes since this field of research is in its beginnings.
On the other hand, a study has pointed out that wild animals living in polluted environments (such as crocodiles) could be a potent source of anti-tumour agents. Since in that milieus, they feed on rotten meat, are exposed to heavy metals, endure high levels of radiation, and are among the very few species to survive the catastrophic Cretaceous-Tertiary extinction event with prolonged lifespan. Therefore, it is reasonable to speculate that they have developed mechanisms to defend themselves against cancer (Animals living in polluted environments are a potential source of anti-tumor molecule(s). https://link.springer.com/article/10.1007/s00280-017-3410-x).
Considering the mentioned comments, several alterations have been made in 4. Future Perspectives and 5. Conclusions sections in order to answer these items and improve overall the quality of the paper.
Comment 7: I was also a bit confused by the discussion of multidrug resistant cancer types. Obviously this is a problem, but surely there is no horizontal transmission of drug resistance similar to antibiotic resistance, and that resistance arises independently within individuals during the course of treatment.
Authors: Thank you for the revision and in order to answer the mentioned problems, 1. Introduction section has been modified (see lines 29-129). As it is mentioned in the text, new strategies to overcome MDR cancer types has been studied (“Collateral sensitivity of natural products in drug-resistant cancer cells. https://www.sciencedirect.com/science/article/pii/S0734975019300096 This information has been completed from the articles “Epoxylathyrane Derivatives as MDR Selective Compounds for Disabling Multidrug Resistance in Cancer https://www.frontiersin.org/articles/10.3389/fphar.2020.00599/full , and “Insights into new mechanisms and models of cancer stem cell multidrug resistance. https://www.sciencedirect.com/science/article/pii/S1044579X19301646” .
This resistance can be a) intrinsic or b) acquired, being the further caused by preexisting factors of the tumor that are present prior to any treatment administered, making consequently certain treatments useless at non-toxic doses. The latter appears to be a relatively common issue throughout the administration of treatment and seems to be the main perpetrator of treatment failure in cancer patients, usually after a relapse.
Comment 8: If so, why is MDR a particular problem now compared to before? Is it because treatments have become more effective so that patients live longer, which gives a larger window of opportunity to develop resistance?
Authors: Thank you for the comments. Considering the cancer a multifactorial disease, the treatment effectivity depends on the internal and external factors of each patient so it can not be related the developing of this disease and the appearance of resistances only to the ageing.
In order to answer this question and also include it in the text to improve the manuscript, the text has been modified (lines 31-66):
“Cancer comprises a group of diseases characterized by anormal cell grown without control, with the potential to invade and to spread through the body [1]. Nowadays, although all the therapeutic options available (surgery, radiotherapy, monotherapy and polytherapy combination regimens that can include the simultaneous use of biological drugs) can reduce tumor size and increase life expectancy [2,3], cancer is still considered one of the main causes of morbidity and mortality worldwide according to the World Health Organization (WHO) [4]. This is due not only to new cases (relapses), but especially because tumours are gained cross resistance to several unrelated chemotherapeutic agents, developing a multidrug resistance (MDR) phenotype, ultimately remain the second cause of death in developed countries [3,5].
Recent evidence points toward stem-like phenotypes in cancer cells, promoted by cancer stem cells (CSCs), as the main culprit of cancer relapse, resistance to radiotherapy, hormone therapy, and/or chemotherapy, and metastasis. Many mechanisms have been proposed for CSC resistance, such as drug efflux through ABC transporters, microenvironment modulation, epigenome, exomes, overactivation of the DNA damage response, apoptosis evasion, increased unfolded protein response (transforming the cells particularly susceptible to Endoplasmic Reticulum Stress and mitochondrial damage), autophagy deregulation, metabolic alterations, prosurvival pathways activation and/or cell cycle promotion [3]. These mechanisms that are still not completely understood and could occur simultaneously [8].Nonetheless, targeted therapy toward these specific CSC mechanisms is only partially effective to prevent or abolish resistance, suggesting underlying additional causes for CSC resilience [3].
This resistance can be a) intrinsic or b) acquired, being the further caused by preexisting factors of the tumor that are present prior to any treatment administered, making consequently certain treatments useless at non-toxic doses. The latter appears to be a relatively common issue throughout the administration of treatment and seems to be the main perpetrator of treatment failure in cancer patients, usually after a relapse. Increasing evidence demonstrates that cells with acquired resistance to a specific drug are prone to exhibit cross-resistance to other chemotherapeutics. This implies the existence of common mechanisms of resistance, which may be independent of the particular action of the chemotherapeutic agent. Importantly, clinical evidence shows that phenotypes of resistance can be reverted to a sensitive phenotype and suggests that cancer associated genetic alterations are not the only players in resistance. Cancer treatments should target not only the resistance mechanisms already present in the bulk of cancer cells but also those activated in CSCs [3].
The influence of genetics and the environment on cell evolution was suggested to affect individual cells at various levels. Modulation of the CSC epigenome, with the acquisition of undifferentiated, pluripotent and drug-resistant phenotypes, occurs preferably when CSCs are exposed to environmental stressors, such as inflammation, toxics (including drugs) and/or radiation [3]. In fact, it may be possible that some nuclear medicine cancer diagnostic and therapeutic techniques favoring the autophagy resulting in development of resistances [6].
(To combat drug resistance, there are two main strategies…)”.
Comment 9: Or it is a byproduct of some other factor, e.g. off-target effects of medicines that are used routinely prior to cancer diagnosis? I think this issue merits a bit more discussion, especially how it relates to the usefulness of animal-based/inspired treatments.
Authors: Thank you for the suggestions that will improve the manuscript. Among the possible causes of MDR, it is possible that some nuclear medicine cancer diagnostic and therapeutic techniques favour the autophagy deregulation (“Insights into new mechanisms and models of cancer stem cell multidrug resistance. https://www.sciencedirect.com/science/article/pii/S1044579X19301646“) . The off-targets effects of medicines could be classified as example of iatrogenic effect. One possible justification of the appearance of resistance using this type of techniques may induce autophagy mechanism as it was mentioned in the article “Autophagy contributes to resistance of tumor cells to ionizing radiationhttps://www.thegreenjournal.com/article/S0167-8140(11)00312-4/fulltext”. However, as neither the beneficial/negative biochemical and physiopathological mechanisms description of autophagy implication in cancer (see paper: “Anticancer Hybrid Combinations: Mechanisms of Action, Implications and Future Perspectives http://www.eurekaselect.com/168866/article”), nor all the MDR mechanisms, promoting agents, implications… are well known and/or described, we do not have enough data to make categorical statements about these issues although this is an interesting and necessary research field.
Therefore, our work is focused on showing the potential opportunities and highlight the importance that animal-based/inspired treatments can offer us in oncotherapy to make an starting point of this area and put it in the spotlight in the near future, but not in deepen in the discussion of MDR resistance or in making assumptions in their specific mechanism when in the majority of the cases have not been well-studied yet.
In view of the comments, several alterations have been made in Introduction section (lines29-129) in order to answer these items and improve the overall quality of the paper.
Considering the Minor comments
Comment 10: Figure 1: It might be useful to include a broken line between humans and animals in the left side of the figure, to indicate how humans might learn to identify active substances by observing local animals species
Authors: As it has been proposed, a broken arrow was included in the figure 1.
Comment 11: Lines 78-80: I'm a bit sceptical that this is the first time this has proposed, unless you specifically mean in the context of MDR cancer
Authors: Yes, it is the first time that a revision gather examples of Zoopharmacognosy and Zootherapy in cancer encompassing them in the term of Zoopharmacology.
In order to improve what it is mentioned, the sentence change from “ In this paper, it is proposed for the first time the Zoopharmacology (comprising Zoopharmacognosy and Zootherapy) as encouraging way to find new cancer treatments that could help to overcome MDR inhibiting drug transporters and/or by new drugs that hypersensitize resistant cancer cells through collateral sensitivity (Figure 2).”
To (now lines 117-120) “In this paper, it is proposed for the first time the Zoopharmacology (comprising Zoopharmacognosy and Zootherapy) as encouraging way to find new cancer treatments in the context of helping to overcome MDR inhibiting drug transporters and/or by new drugs that hypersensitize resistant cancer cells through collateral sensitivity (Figure 2).”
Also, several modifications has been made in Zoopharmacognosy section (lines 131-303) to strengthen the issue defended.
Comment 12: Lines 116-118: In Abbott 2014 (https://onlinelibrary.wiley.com/doi/full/10.1111/een.12110) there is a discussion of the importance of identifying a negative effect of consumption of the substance in the absence of illness, order to determine that it is truly an example of self-medication. It might be worth bringing this issue up in more detail here. Sometimes animals consume specific substances because they need more of a given type of nutrient when ill, but the nutrient does not have a curative effect in itself. Particularly bioactive compounds with a negative effect on cancer are likely to be dangerous to consume in large amounts when unaffected, so this could be an important property to consider when trying to identify potentially useful compounds. (This reference might also be appropriate to cite on lines 94 and 121.)
Authors: Thank you for the comments. The reference has been included as number [18].
Line 94 (now lines 139-142) was completed with: “Although evidence for self-medication in non-human animals was initially mostly anecdotal, increased research in this area over the past two decades has resulted in convincing evidence of self-medication in a number of vertebrates and invertebrates[18].”.
Lines 166-176 were included from the article: “Therefore, the list of the four criteria that must be met to establish self-medication can be summarized in:
- The substance in question must be deliberately contacted.
- The substance must be detrimental, in this case, to tumoral cells.
- The detrimental effect on cancer cells must lead to increased host fitness.
- The substance must have a detrimental effect on the host in the absence of tumoural cells [18].
The latter is the most important factor in separating self-medication from other behaviours. Criterion 4 can be redefined considering the importance of dose-dependence to state that the substance must be detrimental to uninfected individuals when ingested at the level ingested by infected individuals.
Another useful distinction is when the substance is consumed since therapeutic self-medication differs from prophylactic self-medication in that consumption occurs after infection [18].”
Line 121 (now lines 177-178) was completed with: “(…) Huffman [12], Costa-Neto [14], Krief [18],Lozano [17] and Abbott [18] among others.”
Comment 13: Lines 253-255: Sponges seem to have a rather unique immune system that doesn't build on specialized immune cells, but rather excretion of biologically active compounds throughout the organism (i.e. a type of humoral immunity). Such compounds might be more likely to have broad-scope effects than those associated with cell-mediated immunity, such as is common in mammals. Insects also use humoral immunity rather than cell-mediated immunity and antibodies. Perhaps the species with an immune system which is least similar to humans could be of most interest? This could be worth mentioning.
Authors: Thank you for the comments. What it is mentioned is an interesting observation, but, for example, the recent outbreak of COVID-19 it appears to be an example of zoonosis whose possible reservoir are the Pangolins. A recent study shows that these animals lack of a cytoplasmatic RNA sensor that initiates the immune response to the virus which could be a key to understand this disease in humans (Pangolins Lack IFIH1/MDA5, a Cytoplasmic RNA Sensor That Initiates Innate Immune Defense Upon Coronavirus Infection, https://www.frontiersin.org/articles/10.3389/fimmu.2020.00939/full ). Moreover, it appears llamas develop antibodies that could effectively fight SARS-CoV-2 infection (https://www.sciencefocus.com/news/could-antibodies-found-in-llamas-help-us-to-defeat-covid-19/ ). These antibodies are present in these animals but not in humans and this research group work with llamas because they generate a different immune response than humans (although both are mammals). Also, there are other points to consider such as their feeding (that pangolins are insectivorous while llamas are herbivorous), their environment…
Therefore, since the rationalization of this type of research with different type of animals is in its beginnings in the majority of diseases (which also includes cancer), it may be a bit hasty mentioning it.
Additional authors notes for the reviewer:
- The sentence “The overlapped similarities between the ingestion of plants by primates and their medicinal use by humans provides and could warrant a bio-rational for the search of bioactive plant in the primate diet with pharmacological and phytochemical value [41].” place changes from lines 166-168 to 232-234.
- Table 1 format (lines 360-363) has been re-adapted to the new version of the manuscript.
- It was created a new section, Section 4 title change from 4. Conclusions to 4. Future perspectives.
- Conclusions is now section 5.

Round 2
Reviewer 2 Report
I'm happy with the new additions to the manuscript in response to my previous comments. I think the authors have been very successful in their revisions, and have no further comments.